# Bioabsorbable Osteofixation Materials for Maxillofacial Bone Surgery: A Review on Polymers and Magnesium-Based Materials

**DOI:** 10.3390/biomedicines8090300

**Published:** 2020-08-21

**Authors:** Sung-Woon On, Seoung-Won Cho, Soo-Hwan Byun, Byoung-Eun Yang

**Affiliations:** 1Division of Oral and Maxillofacial Surgery, Department of Dentistry, Hallym University Dongtan Sacred Heart Hospital, Hwaseong 18450, Korea; drummer0908@hanmail.net; 2Graduated School of Clinical Dentistry, Hallym University, Chuncheon 24252, Korea; kotneicho@gmail.com (S.-W.C.); purheit@daum.net (S.-H.B.); 3Institute of Clinical Dentistry, Hallym University, Chuncheon 24252, Korea; 4Division of Oral and Maxillofacial Surgery, Hallym University Sacred Heart Hospital, Anyang 14066, Korea

**Keywords:** bioabsorbable plate, biodegradation, biocompatibility, polymer, magnesium, maxillofacial osteofixation, corrosion

## Abstract

Clinical application of osteofixation materials is essential in performing maxillofacial surgeries requiring rigid fixation of bone such as trauma surgery, orthognathic surgery, and skeletal reconstruction. In addition to the use of titanium plates and screws, clinical applications and attempts using bioabsorbable materials for osteofixation surgery are increasing with demands to avoid secondary surgery for the removal of plates and screws. Synthetic polymeric plates and screws were developed, reaching satisfactory physical properties comparable to those made with titanium. Although these polymeric materials are actively used in clinical practice, there remain some limitations to be improved. Due to questionable physical strength and cumbersome molding procedures, interests in resorbable metal materials for osteofixation emerged. Magnesium (Mg) gained attention again in the last decade as a new metallic alternative, and numerous animal studies to evaluate the possibility of clinical application of Mg-based materials are being conducted. Thanks to these researches and studies, vascular application of Mg-based biomaterials was successful; however, further studies are required for the clinical application of Mg-based biomaterials for osteofixation, especially in the facial skeleton. The review provides an overview of bioabsorbable osteofixation materials in maxillofacial bone surgery from polymer to Mg.

## 1. Introduction

Clinical application of plates and screws for osteofixation in maxillofacial bone surgery is an essential element for surgery requiring rigid fixation. For the past few decades, titanium was the material of choice to manufacture plates and screws for rigid fixation in the maxillofacial area through the accumulation of many clinical studies and clinical experience. Titanium-based screws and plates show sufficient strength and Young’s modulus values for rigid fixation during the healing process of the fractured bone, and they are biologically inert. Moreover, titanium can bind to the bone and can be maintained asymptomatically in most cases [1]. However, there are some limitations and risks associated with titanium plates. Growth disturbance in children, interference with radiological imaging, hypersensitivity to cold stimulation, and stress shielding may result from the application and maintenance of titanium devices [2,3,4,5,6]. Due to these limitations, titanium plates and screws are often removed through second operations at the request of the patient. Furthermore, the need for second surgeries is time- and cost-consuming. Consequently, the demand and interest for absorbable plates and screws resulted in various experiments and researches to develop biodegradable osteofixation devices.

Polymer-based osteosynthesis materials such as polyglycolic acid (PGA), poly-l-lactic acid (PLLA), poly-d-lactic acid (PDLA), copolymers of PGA, PLLA, and PDLA, and unsintered hydroxyapatite (u-HA)/PLLA were introduced to overcome concerns about titanium plating devices [7,8,9]. They could degrade spontaneously after healing of the fractured bone segment. Therefore, the secondary surgery to remove the plates and screws was no longer needed. These polymer-based materials also provided some advantages over titanium-based osteofixation, which consist of the absence of metal corrosion, no interference with a radiological evaluation due to their radiolucent properties, and reduced stress-shielding effect [10,11,12]. However, due to concerns about their questionable physical strength and cumbersome procedures for bending and fitting the plates, a desire for the new materials presenting the advantageous traits of metal plates and screws increased. As a result, magnesium (Mg) was recently highlighted as new material. Mg alloys were already developed as degradable metallic materials for orthopedic and vascular applications in the late 1800s [13]; however, they lost attention due to excessive hydrogen gas formation and early loss of mechanical strength resulting from rapid corrosion. In recent years, they were drawn in the limelight again with the help of numerous studies solving these problems and with verification of biocompatibility. The purpose of the present article is to provide an overview of bioabsorbable polymer- and Mg-based materials for osteofixation in maxillofacial bone surgery.

## 2. Conventional Bioabsorbable Plates and Screws

As mentioned above, PGA, PLLA, copolymers of PGA, PLLA, and PDLA, and u-HA/PLLA are classified in this category. Polymer-based bioabsorbable materials evolved since the first attempt to use bioabsorbable materials in the maxillofacial area to stabilize the fracture of the facial skeleton in 1971 [14]. Initially, the low strength of the absorbable plates and screws resulted in a solid and large form to provide sufficient mechanical stability. However, advances in self-reinforcing technology [15] enabled the manufacture of bioabsorbable devices with higher strength and durability, as well as improved biocompatibility. The mechanism of self-reinforcement is paralleling polymeric fibers and blending polymers to make the oriented structure. After self-reinforcement, the mechanical strength, modulus, and toughness of polymers increase significantly. This self-reinforced (SR) biodegradable plates and screws showed suitable strength and adequate stability to fix fractures in areas with high loads, such as the mandible [16,17,18,19]. As a result, polymer-based bioabsorbable plates and screws were applied in maxillofacial trauma surgery, orthognathic surgery, and pediatric surgery (Table 1).

### 2.1. Characteristics of Polymer-Based Bioabsorbable Materials for the Plate and the Screw

#### 2.1.1. PGA

PGA was the first bioabsorbable polymer developed for pins, screws, and plates in bone surgery. It is a hard and tough polymer with an average molecular weight of 2.0 × 10^4^ to 1.45 × 10^5^, melting at about 224 °C, and it has a highly crystalline structure. Although its clearance time is 6–12 months, due to its property of rapid degradation, PGA shows early loss of mechanical strength in vivo, approximately 4–7 weeks after implantation [9,20]. Additionally, osteofixation using PGA was associated with adverse effects such as swelling, fluid accumulation, and sinus formation when used in orthopedic surgery [21]. For this reason, only a few cases of maxillofacial surgeries were performed using PGA [9,20].

#### 2.1.2. Poly-Lactic Acid (PLA): PLLA and PDLA

PLA is another bioabsorbable polymer with a high molecular weight (1.8 × 10^5^ to 5.3 × 10^5^). It has two stereoisomeric forms as a result of the optically active carbon in lactate: PLLA and PDLA. PLLA is a pale polymer, and its melting point is about 174 °C to 184 °C. PLLA is regarded as the first generation of bioabsorbable osteosynthetic material, and it was used in maxillofacial surgery since the early 1990s. It shows hydrophobicity and, therefore, is more resistant to hydrolysis than PGA.

PLLA is known to show a long degradation time—about two years in vitro [9,22] and over 3.5 years in a clinical setting. Its reported problems were foreign body reactions and adverse tissue response due to a slow degradation when used for fixation in the case of zygomatic fracture [23]. In contrast to PLLA, PDLA has lower crystallinity and shows lower resistance to hydrolysis. However, because PDLA also has a relatively long degradation period, its crystalline particles may cause some inflammatory response [24]. There is no study using pure PDLA for osteofixation in the maxillofacial area.

#### 2.1.3. Copolymers of PGA, PLLA, and PDLA

Copolymers of PGA, PLLA, and PDLA are called the second generation of bioabsorbable osteosynthetic material, and they exhibit faster absorption rates than the first-generation polymers. Using copolymerization of different derivatives of α-hydroxy acids, various mechanical properties and degradation rates can be obtained. In general, the more glycolide content there is, the faster the degradation rate it shows. The byproducts of this copolymer via the citric acid cycle are carbon dioxide and water, which are excreted by the lung [25,26] (Figure 1). This means that copolymers do not yield any crystalline particles in contrast to pure polymer. Furthermore, the degradation rate of the copolymers is slow enough to achieve high biocompatibility [8,9]. Most devices that consist of these copolymers are usually made of poly-l-*co*-dl-lactide (PLDLA) copolymers and copolymers of PLLA and glycolic acid (PLGA). PLDLA copolymers have similar mechanical properties compared to PLA but have a high biodegradation rate. In the case of SR-PLDLA, it takes 2–3 years for absorption. PLGA was developed to achieve early controlled absorption, as well as stability, and it provides adequate strength for 6–8 weeks with a reasonable resorption time of 9–15 months [27,28,29].

#### 2.1.4. Unsintered Hydroxyapatite (u-HA)/PLLA

Another type of conventional bioabsorbable material is a u-HA/PLLA composite, which consists of fine particles of u-HA and carbonate ion combined with PLLA. In the case of sintered HA, although its absorbability is sometimes reported, it is not completely absorbable. Therefore, it is not considered as intrinsically bioabsorbable ceramics, and it cannot be used in manufacturing absorbable plates and screws for internal bone fixation [30]. The purpose of incorporation of hydroxyapatite into the polymer is to obtain osteoconductive capacity. As u-HA/PLLA composites can be osteoconductive and biodegradable, there is a high possibility that they can be replaced entirely by bony tissue [30]. These composites are considered the third generation of bioabsorbable osteosynthetic material. The u-HA/PLLA composite materials show higher mechanical strength, such as bending strength, shear strength, and impact strength, than PLLA-based materials. In addition, they have a radiopacity that enables follow-up after surgery. Due to this advantage, they can be clinically advantageous, and they are expected to be used frequently in various maxillofacial surgeries.

### 2.2. Clinical Application

The maxillofacial area, especially midfacial skeletal area, is considered to be a suitable site for the application of bioabsorbable osteosynthetic devices because fractures in this region are relatively easy to access while the biomechanical stress exposed to this region is relatively low.

As the first clinical study on the use of PGA for fracture of the facial skeleton, Roed-Peterson [31] in 1974 reported acceptable results after a six-week fixation period of mandibular angle fractures in two pediatric patients. However, the PGA used in this study was not in the form of a plate, but merely a suture. As mentioned earlier, PGA loses its mechanical strength in six weeks and can induce various adverse tissue responses [21]. Due to these limitations, PGA was rarely used in the maxillofacial area, although it was previously applied in the orthopedic field. There is no study on the use of plates and screws made by PGA alone in maxillofacial surgery.

Interests in PGA as an absorbable material shifted to PLA, which shows a slower degradation rate. Bos et al. [32] in 1987 demonstrated that fixation using PLLA plates and screws showed successful bone union in 10 patients with zygomatic fracture. Gerlach [33] in 1993 reported the successful application of PLLA osteosynthesis material for the treatment of zygomatic and mandibular fractures in an in vivo and clinical pilot study. In 1997, Bessho et al. [34] applied PLLA plates and screws to 50 patients with fracture of zygomatic bone or maxilla or mandible. They reported adequate bone healing, except for the two patients with postoperative infection. However, another study reported delayed subcutaneous swelling at the site of implantation associated with the material [23]. Therefore, the clinical application of pure PLA in the maxillofacial area was not achieved sufficiently. Pure PLA-based materials were never applied in orthognathic surgery. As a unique study that should be mentioned among researches using PLLA plates, Tams et al.’s study [35] can be acknowledged. In their investigation, PLLA plates and screws were applied after mandibular swing osteotomy for cancer ablation [35]. They evaluated the healing of the bone in four patients who had PLLA plates for osteofixation following mandibular swing osteotomy and observed that uneventful bone healing was achieved in all patients, except for one in whom callus formation was observed. Histological examination performed after 5.5 years showed a non-specific foreign body reaction to the PLLA remnant, which was still present. As PLLA does not cause changes in the local dose distribution in radiotherapy [36], Tams et al. [35] demonstrated that the application of biodegradable implants should be considered in patients with malignant tumors when postoperative radiotherapy is required.

Meanwhile, as the clinical use of rods or screws made of self-reinforced PGA and PLLA in orthopedics was prevalent from the late 1980s to the early 1990s, experiments on animals were conducted to apply SR-PLLA in the maxillofacial area [37,38]. At that time, the fixation of fractures using rods or screws was not commonly performed in maxillofacial surgeries. However, the use of lag screws for the mandibular fracture by Ellis and Ghali [39,40] provoked the use of screws for osteofixation in the maxillofacial region. The change was also made with bilateral sagittal split osteotomy (BSSO). Suuronen et al. [41] applied SR-PLLA screws to eight patients requiring mandibular advancement or retraction, and uneventful primary healing during the follow-up period ranging from 15 to 23 months was reported. In their study, only two patients had a light maxillomandibular fixation with elastic bands for 15 to 25 days, but the rest did not. Since then, many studies were conducted to apply SR-PLLA to orthognathic surgery. In a study that applied SR-PLLA screws to BSSO surgery in 25 patients for the mandibular advancement, the biodegradable SR-PLLA screws were reported to be comparable to other types of materials for rigid fixation in terms of postoperative skeletal stability [42].

Le Fort I (LF I) osteotomy is challenging to fix using a lag screw or positional screw due to the structure of the maxilla, and plates must be used for osteofixation. In 1998, Haers et al. [11] published a pilot case using plates and screws made of SR-PLDLA copolymer for bimaxillary orthognathic surgery with genioplasty. Although the follow-up period was a short six weeks, healing was achieved without event, and postoperative occlusion was stable. Dental and skeletal relapse of the maxilla and mandible was not detected except for 0.5 mm of the relapse tendency of Menton point. Subsequently, in 2002, Turvey et al. [43] performed orthognathic surgery using plates and screws made of SR-PLDLA in 70 patients to reposition the maxilla and/or mandible. Although immediate loosening of the screw occurred in three patients (one patient required emergent reintubation due to laryngospasm, and the other two patients with Tourette syndrome and Down syndrome, respectively, showed loosening of fixation due to postoperative facial tics or behavior), the remaining 67 patients showed favorable healing and no short-term problems. SR-PLDLA was also applied in the field of maxillofacial trauma. Yerit et al. [18] managed 22 patients with various mandibular fracture patterns using the SR-PLDLA device and reported that, during approximately 49 weeks of follow-up, mucosal healing and fracture consolidation were normal in 20 patients, while mucosal dehiscence occurred above the absorbable device in two patients. They suggested that SR-bioabsorbable osteosynthesis materials could be a reliable alternative to titanium plate systems. Ylikontiola et al. [19] also used SR-PLDLA-based plates and screws for the fixation of the mandibular fracture. However, their study differs from the Yerit et al.’s study in that they applied to an isolated anterior mandibular fracture, including paraymphyseal fracture. They concluded that the SR-PLDLA device is reliable for internal fixation of the anterior mandibular fracture. However, adequate soft tissue management should be required to prevent plate exposure and infection. When it comes to applying SR-PLDLA to children, another study by Yerit et al. in 2005 [44] can be exemplified. They applied SR-PLDLA to children with 13 mandibular fractures and reported that, during 26.4 months of follow-up on average, primary healing was observed in all patients, while significant complications such as adverse tissue reaction, malocclusion, and growth inhibition were not observed. In fields other than maxillofacial trauma and orthognathic surgery, a study using SR-PLDLA in an ablative oral cancer surgery was reported. Ketola-Kinnula et al. [45] reported 15 cases where SR-PLDLA plates and screws were used for fixation of mandibulotomy in ablative cancer surgery. There was no particular problem at the time of surgery, but one patient required reoperation due to the failure of fixation, which was caused by osteoradionecrosis and fracture of the plate. Furthermore, during follow-up (median 3.5 years), six cases showing nonunion were detected radiologically, although three of them were clinically stable. Based on the results of their studies, Ketola-Kinnula et al. concluded that bioabsorbable plates and screws should not yet be used for fixation of osteotomy in cancer surgery.

Since PLGA was approved by the United States (US) Food and Drug Administration (FDA) for craniomaxillofacial fixation in 1996, studies using PLGA as a fixation material were also conducted. Edwards and Kiely [46] performed LF I osteotomy using PLGA plates and screws on 29 patients, and they evaluated the fixation devices in terms of wound healing, the stability of fixation, infection signs, and patient satisfaction. Complications were not observed except for one case where the L-shaped plate was touched in the paranasal area. They demonstrated that the PLGA plates and screws showed favorable results and could be an additional option for the fixation of the maxilla. Subsequently, studies using PLGA plates and screws as a resorbable plating system not only in LF I but also in BSSO were reported. In the study of Shand and Heggie [47], 31 patients underwent routine orthognathic procedures and recovered normally, except for one patient who developed buccal space abscess. They concluded that surgical technique was required for the success of the surgery, but claimed that PLGA was still a good fixation material for repositioning the maxilla and mandible.

Additionally, there are some studies that applied PLGA to trauma surgery. Landes et al. [48] applied PLGA and PLDLA to 80 patients for fixation of maxillary and mandibular fracture or reconstructive procedures for correction of dysgnathia or hemifacial macrosomia. In their study, PLGA was used in 20 patients, one of whom showed a reaction requiring curettage. They also reported that PLGA degraded within 12 months, while PLDLA degraded within 24 months, and that both copolymers showed reliable biocompatibility and disintegration. In a study that applied three bioabsorbable devices to the zygomatic fracture [49], PLGA was used in 18 patients, and it was reported that it could be used without problems in the frontozygomatic suture and paranasal regions except for the infraorbital rim and zygomaticomaxillary crest/anterior sinus wall. Among the more recent literature, there are studies in which PLGA was applied to pediatric mandibular fractures. In these studies, it was reported the use of PLGA in pediatric mandibular fractures is an effective and safe method for internal fixation [50,51].

In terms of uHA/PLLA, which was introduced with the expectation of osteoconductive capacity, it showed good clinical results. In orthognathic surgery, the clinical usefulness of uHA/PLLA was confirmed through its application in BSSO by Ueki et al. [52] and in bimaxillary surgery by Landes et al. [53]. As far as trauma is concerned, adequate strength and few complications of uHA/PLLA used for facial fractures, including malar and midfacial fractures, as well as orbital floor and medial wall fractures, were reported [54,55,56,57]. For pediatric surgery, it was reported that, when applied with dental stabilization using an arch bar or orthodontic wire to mandibular fractures, it can be applied to displaced mandibular fractures in children without complications such as damage to tooth buds, postoperative infection, malunion, and nonunion [58].

Finally, it is safe to conclude that the use of polymer-based bioabsorbable materials in maxillofacial fractures and orthognathic surgery might be a good treatment option when indications are screened appropriately, although it cannot be said which polymer is better. Some studies compared several polymer-based materials in maxillofacial surgery [48,49,59,60], but the authors of these retrospective studies did not conclude which material was better. To the best of our knowledge, no randomized controlled trials compared polymer-based bioabsorbable materials in the maxillofacial area. It is believed that further studies, especially randomized controlled trials, are needed to determine which polymer is better.

### 2.3. Limitations

Although polymer materials were developed to overcome the problems associated with metal plate systems, they are also not ideal materials, and their limitations were reported. Such limitations include concerns about their physical strength, the hassle of bending and molding process of plates, and radiolucent images that interfere with postoperative monitoring. Specific complications of absorbable plates and screws include wound dehiscence, plate exposure, malunion, nonunion, and foreign body reaction. Foreign body reactions were reported with local swelling, sterile abscess, fistula formation, and osteolysis [16,61,62,63], and the most common of these was aseptic abscess [61]. According to a meta-analysis of the complications of the absorbable plate system in maxillofacial surgery in 2013 [12], foreign body reactions and plate mobility were significantly more frequent than those using titanium. It was concluded that, in maxillofacial surgery, the absorbable plates and screws do not show adequate safety profiles. A systematic review of the application of absorbable fixation devices in pediatric craniomaxillofacial trauma reported overall major and minor complications from 3.21% to 5.45%, and malocclusion, extrusion, infection, fistula formation, and hypoesthesia were the most common complications [64]. One of the advantages of the absorbable plate system is that no secondary operation is necessary. However, recently, in a randomized controlled trial [65] comparing the absorbable plates and screws with those of titanium-based system, the necessity of removing the absorbable plates and screws due to complications such as infection was found to be twice as high as that of titanium. It was demonstrated that the possibility of intraoperative switches due to the material failure of the bioabsorbable system also reduced the justification of the use of absorbable plates and screws in maxillofacial surgery [65]. Furthermore, in another meta-analysis, it was reported that perioperative screw breakage was more frequent in the bioabsorbable osteosynthesis group than the titanium osteosynthesis group [66]. Eventually, due to these limitations and complications of polymer-based absorbable osteofixation materials, there is a desire for materials with higher strength while having the advantages of absorbable osteosynthesis, and Mg is in the spotlight as a material which can satisfy such conditions.

## 3. New Bioabsorbable Plates and Screws—Magnesium Plates and Screws

As a new material for plates and screws, Mg is attracting attention to replace polymer-based materials. Magnesium is a widely distributed element in nature. It is an element that is the second most abundant in the hydrosphere and the eighth most abundant in the lithosphere. Due to its high level of reactivity as a free element, Mg is only found in the form of divalent cations (Mg^2+^) or salts or minerals throughout the biosphere [67]. It is noteworthy that Mg plays an essential role in all reactions requiring chlorophyll molecules and adenosine triphosphate [68,69]. Because Mg is also the fourth most abundant element found in the human body and the most abundant intracellular divalent cation, it is involved in more than 300 known enzyme reactions [67,70]. In addition to many other cellular functions, Mg plays a role in the synthesis of protein and nucleic acid, mitochondrial activity and integrity, modulation of ion, and stabilization and translation processes of the plasma membrane [68,70,71,72]. There is a potential of disrupting Mg balance in the human body, especially when the excess of Mg results from use as an osteofixation material. In other words, the use of Mg-based biomaterials may result in excess circulating and stored Mg, which could clinically manifest as hypermagnesemia. Early symptoms of hypermagnesemia involve a drop in blood pressure, mental impairment, and nausea [73]. Higher concentrations of circulating Mg are related to the impairment of the neuromuscular system, and progressive muscle weakness can lead to respiratory failure [74]. Therefore, hypermagnesemia conditions should be avoided. Although the body can cope with a wide range of Mg concentrations under normal renal capacity, slowing material corrosion is the best way to prevent hypermagnesemia when considering use in the human body. Therefore, research is ongoing to slow down the corrosion of Mg and investigate its biocompatibility in the human body.

### 3.1. History of Mg as a Biomaterial

Shortly after the commercialization of Mg production in the mid-19th century, metals were first used as biomaterials. Edward C. Huse is known to be the first to use Mg as a wire ligature to prevent bleeding during surgery in three patients in 1878 [75]. Over the next 50 years, numerous physicians tried applying Mg and Mg alloy devices to vascular, orthopedic, and general surgery. Among the early pioneers, Erwin Payr primarily led research promoting Mg as a biomaterial through a series of investigations in the early 20th century [76]. He applied Mg to a wide range of operations, the most effective of which was the application for hemangioma treatment and the suturing of organs rather than use in the field of orthopedic surgery. Moreover, relatively successful applications such as clips for vascular anastomosis, clips for gastrointestinal anastomosis, and tubes used in ureterorectostomy were reported [75,76,77]. However, these applications focused only on rapid absorption rather than mechanical integrity; thus, the effect of hydrogen gas production on tissue damage could not be well evaluated.

The application of Mg for orthopedic use showed slightly more varied results. Among researchers who initially applied Mg to the musculoskeletal system, Albin Lambotte is worth mentioning in this paper. Through his earliest attempt to use Mg plate with steel screws to fix the fractures of the tibia and fibula in a 17-year-old boy in 1907 [78], which failed due to galvanic corrosion, he realized that two different types of metallic materials should not be used together. Subsequently, Lambotte conducted a series of animal studies to reveal that, when used alone, Mg implants partially corroded after three months and completely corroded after 7–10 months [78]. Jean Verbrugge, Lambotte’s assistant, continued this study to investigate Mg implants in both animals and humans, and he reported the successful use of Mg with no adverse reactions, although it caused a small volume of harmless gas [76]. On the contrary, Ernest Hey Groves [79,80] investigated the application of Mg intramedullary pegs in experiments on animals in 1912–1913. He reported excessive callus formation and rapid corrosion of the implant, which disintegrated adequate healing. In the late 1930s, Earl McBride [81] used screws and nails made of Mg alloys containing aluminum and manganese for fixation of fractures in humans. He made some significant observations, including that, because pure Mg could not provide the required strength, alloys were preferred in this purpose, and that Mg-based implants should not be applied to regions under high mechanical load. Additionally, he confirmed that Mg should be used only when the merits of corrosive materials outweigh the effect on the surrounding tissue.

In the following decades, interest in Mg decreased, presumably due to inconsistent results associated with the rapid corrosion of Mg implants. Mg biomaterials began to gain popularity again in the late 1990s, growing exponentially since then. In the maxillofacial area, animal experiments are underway to develop Mg-based plates and screws.

### 3.2. Science of the Corrosion Behavior of Mg

The rapid corrosion of Mg and Mg alloys is a significant limitation regarding the use of these materials in various applications exposed to a corrosive environment [82]. Figure 2 shows corrosion mechanisms that occur when Mg is exposed to an aqueous environment. As a result of Mg corrosion, magnesium hydroxide and hydrogen gas are produced. When exposed to high chloride concentrations as seen in physiological environments, Mg(OH)_2_ reacts with chloride ions, resulting in the production of highly soluble MgCl_2_. As a result, rapid dissolution of the Mg substrate is achieved, and hydrogen gas and hydroxide ions are produced [83].

There are two main types of corrosion that affect Mg and Mg alloys in physiological environments. In the case of single-phase materials, corrosion is typically localized, forming pits on the surface of the material; however, in the presence of a secondary phase due to impurities or alloying components, it causes galvanic corrosion, and the second phase causes local corrosion as a local cathode [82]. The absence of generalized corrosion is an essential factor in the use of Mg as a biomaterial since the presence of a wide range of corrosion areas is likely to cause mechanical failure of the implant in certain areas. Furthermore, the use of Mg biomaterials that show rapid corrosion due to the corrosion mechanism can result in the production of hydrogen gas within the implant environment, as well as an increase in local pH, which can significantly affect the surrounding tissue. These are the biggest obstacles for successful clinical application of Mg-based materials as osteofixation biomaterials.

### 3.3. Methods for Improving the Performance of Mg for Biomedical Application

Various techniques are currently being studied to improve the appropriate mechanical properties and corrosion resistance of Mg as a biomaterial, along with biocompatibility and potential osseointegration ability. These techniques mainly consist of alloying of Mg, surface treatment, or coating techniques [58,84,85].

Alloying of the metals can change their strength, ductility, and corrosion properties based on each specific composition [86,87]. When it comes to Mg alloys, changing the microstructural characteristics is of primary concern as a method for improving strength and corrosion properties. Hence, attention is focused mainly on reducing particle size compared to pure Mg. In fact, most studies to improve the properties of Mg alloys were conducted primarily for industrial purposes [88]. However, a considerable amount of research conducted initially for the automotive industry turned its purpose toward developing Mg alloys as biomaterials [89]. Other alloys such as aluminum or rare earth elements and are still being investigated for medical applications, but those containing non-toxic elements are also being developed. Mg alloys containing non-toxic elements such as calcium, manganese, zinc, and zirconium can be demonstrated as an example.

Aluminum is commonly added to Mg alloys to improve both the strength and the corrosion resistance [90]. Although the mechanism of improving corrosion resistance is not entirely understood, it seems to be established that increasing concentration of aluminum reduces the corrosion [91,92]. However, there is concern regarding several pathological conditions such as dementia and Alzheimer’s disease associated with aluminum. Since the long-term effects of implanting aluminum-containing materials are not yet identified, it may not be wise to investigate these alloys as biomaterials further before more significant longitudinal studies are conducted [93]. Rare earth elements consist of 15 elements between lanthanum and lutetium in the periodic table, as well as scandium and yttrium. These are commonly added to Mg alloys as hardeners or master alloys used to increase the strength, ductility, corrosion resistance, and creep resistance [89,94]. However, since their physiological effects are not well known, more studies are needed on the biocompatibility of these alloying elements. The addition of calcium has the advantage of increasing mechanical strength compared to pure Mg by reducing the grain size of the alloy. Calcium was widely investigated as part of a binary Mg alloy or in combination with other elements for the application as osteofixative biomaterials. Manganese is used as a component of Mg alloys to improve corrosion resistance via reduction of the detrimental effects of impurities such as iron [82,95]. Furthermore, enhancement of the ductility and improvement of the yield strength of Mg alloys can be obtained via the addition of manganese [89,96]. Zinc can improve the strength of the Mg alloy, and, similar to manganese, it can reduce the corrosion-enhancing effects of common impurities such as iron, nickel, and copper [82]. However, since cytotoxicity was identified in vitro when cells were exposed to high concentrations of zinc [97,98], potential toxicity could not be excluded entirely. Zirconium has a highly effective grain-refining ability. The addition of zirconium to Mg alloys improves the strength of the alloy and the corrosion resistance via precipitation of combined zirconium–iron particles before the alloy is cast [82,99]. Zirconium is already commonly used in the field of medical implants, including dental and orthopedic implants, and it is widely accepted that zirconium shows biocompatibility.

In addition to the components above, elements such as lithium, cadmium, tin, strontium, silicon, silver, and bismuth are also being investigated. These alloys can be binary, ternary, or more, and the components and composition of the alloys significantly contribute to different mechanical properties and corrosion behavior [13]. The addition of lithium to Mg alloys can enhance ductility by lowering the density of Mg and improving low-temperature toughness [100]. Lithium was approved by the US FDA to treat bipolar and depressive disorders [101,102]. According to several reports, lithium ions can promote bone formation and enhance bone density in vivo [103,104,105]. It is thought that lithium could be promising as a component of Mg alloy. The effect of cadmium in Mg alloy is to improve corrosion-resistant behavior [106]. Furthermore, the addition of cadmium is known to reduce the susceptibility of alloys to stress corrosion cracking [106]. Although the possibility of cadmium toxicity remains, zinc and Mg can reverse cadmium-induced renal toxicity [107].

Surface treatments or coatings are methods of using substrates to improve the biocompatibility of the material in the biomedical field [108]. In order to develop Mg for biomedical use, such surface modifications were mainly investigated to improve the corrosion resistance of the substrate. According to the definition of Wang et al. [109], surface modification methods include chemical surface modification, physical surface modification, or a combination of the two. Chemical surface modifications consist of acid etching, anodization, alkaline treatment, ion implantation, and fluoride treatment. All of these methods involve the replacement of the corrosion-resistant oxide layer on the surface of Mg [109]. The physical coating is a method involving making protective coatings ranging from organic to inorganic or metallic, forming a physical barrier between the metal and corrosive environments, using various techniques [110,111,112]. As a combination method of chemical and physical surface modifications, a method of performing the initial chemical pretreatment to improve the adhesion of the physical coating followed by performing the physical coating was studied [109]. The most widely investigated surface modification technique in the field of Mg biomaterials is calcium phosphate coatings with or without pretreatment. Calcium phosphate shows high biocompatibility and is a crucial precursor for bone growth. It is widely applied in orthopedic and dental fields, and there is some evidence that calcium phosphate coating can increase the corrosion resistance of Mg-based materials [113,114,115,116,117]. However, calcium phosphate coatings are challenging due to improper control of phase formation, cracks, and poor adhesion [111]. Nevertheless, the various surface modification techniques mentioned above show the potential to modulate the corrosion of Mg and Mg alloys for clinical applications as osteofixative materials.

### 3.4. In Vitro and In Vivo Studies for Assessing the Corrosion and the Biocompatibility of Magnesium

Research on the application of magnesium in the maxillofacial field started intensively since 2010. Currently, there is only one case report in which Mg screws were applied to the human body in the maxillofacial region. However, other in vitro and in vivo studies were conducted. These studies focused on the assessment of the corrosion or biocompatibility of Mg-based materials. Due to the rather common interest of dentistry and orthopedic surgery for hard tissues, various studies in maxillofacial surgery are underway based on the favorable results of in vitro experiments in orthopedic surgery, including Mg-based orthopedic screws. In the field of maxillofacial surgery, the biocompatibility of Mg-based materials is being investigated via application of specific Mg alloys in which elements enhancing corrosion resistance such as aluminum, yttrium, manganese, zinc, and zirconium are added, or via additionally applying surface modification to these Mg alloys. Magnesium alloys used for in vivo studies in maxillofacial surgery are presented in Table 2.

In 2014, a study was conducted in which biomechanical evaluation of the Mg-based screw system in BSSO was performed using three-dimensional finite element analysis [121]. Lee et al. [121] created a virtual BSSO model using image processing software capable of three-dimensional design and modeling, followed by fixation using Mg-, titanium-, and polymer-based bicortical screws. Mg screws showed better biomechanical stability than polymer-based screws and biomechanical stability close to titanium screws; thus, they concluded that Mg-based absorbable screws are promising alternatives to polymer-based systems. In vivo studies on the Mg-based osteofixation system also began to overflow. In 2016, Schaller et al. [122] evaluated in vivo degradation of Mg plates and screws using calvarial models of miniature pigs. After a plating system made of Mg alloy WE43 with or without plasma, the electrolytic surface coating was applied to the miniature forehead, while radiographic analyses using X-ray, CT, and μ-CT and histological examination were performed. As a result, subcutaneous gas cavity formation was mainly observed with the uncoated plate, and the coated screw showed increased bone density and bone-implant contact, as well as a lower corrosion rate compared to the uncoated screw, and it was suggested that further research on Mg coating to prevent gas formation is needed for application in the facial skeleton [122]. In addition, studies on the application of Mg for osteosynthesis of cranioplasty in miniature pigs and osteosynthesis using Mg plates and screws in partial rib osteotomy of miniature pigs for the evaluation of cyclic deformation were also conducted, and suitable and promising results for internal fixation were reported [123,124]. As the most recent study using miniature pigs, Naujokat et al. [125] used Mg alloys with surface modification via hydrogenation or fluoridation in the mandibular osteotomy model and compared them with Mg alloys without surface modification. In their study, Mg-based plates and screws with surface modification showed full biocompatibility in soft tissues. Furthermore, the quantified degree of corrosion was lower in both hydrogenated and fluoridated plates compared to plates without surface modification. Healing of the osteotomy sites was uneventful regardless of surface modification, and they concluded that surface-modified Mg plates and screws had no significant effect on bone healing, biocompatibility, and corrosion kinetics when used for osteosynthesis of the mandibular angle. As a unique study using rats, Lim et al. [126] applied hydroxyapatite-coated Mg plates to the frontal bones of Sprague-Dawley rats and compared them with bare Mg plates. Through clinical, hematological, and radiological evaluation using μ-CT, it was reported that the hydroxyapatite-coated Mg plate can delay the absorption by controlling the initial absorption rate, thereby reducing complications such as the rapid formation of hydrogen gas and wound dehiscence [126].

As a preliminary step for more practical application in the maxillofacial area, several studies using beagle dogs were also conducted. Kim et al. [127] performed osteofixation using Mg-based plates and screws for the zygomatic arch fracture of beagles in 2018, and the Mg-based implant showed stronger load and stiffness compared to the control group using absorbable polymer. Although the Mg group showed a temporary increase in hydrogen gas starting from three days post-operation, it decreased spontaneously and did not affect bone healing. In 2020, Byun et al. [118,119] published studies of applying different Mg alloys to the LF I canine model of beagle dogs. Firstly, in the study using the WE43 (Mg, 3.78 wt.% Y, 2.13 wt.% Nd, and 0.46 wt.% Zr) plate and screw [118], the WE43 group pre-treated with an extrusion process showed increased mechanical strength, and the strength was sufficient for application in the midfacial area. In terms of biocompatibility, although gas formation and swelling occurred in the early stage, it resolved over time and showed favorable clinical results. Secondly, they applied PLLA coating to the ZK60 (Mg, 4.8–6.2 wt.% Zn, 0.64 wt.% Zr, 0.012 wt.% Mn, 0.0021 wt.% Fe, 0.0016 wt.% Si, 0.0014 wt.% Al, 0.001 wt.% Cu) plate and used it in the LF I canine model of beagle dog [119]. Although ZK60 has stronger mechanical strength than WE43, it has the disadvantage of showing fast absorption [128,129] Accordingly, PLLA coating was applied to the plate to delay the absorption of plates and screws. However, because the PLLA coating was damaged throughout the bending process, it was not possible to prevent the rapid absorption of ZK60. Thus, they concluded that further studies using different coating materials or different Mg alloys were needed. Uniquely, there is a study in which Mg alloy was applied to the cranial bone of sheep. Torroni et al. [130] evaluated the biological behavior of WE43 alloy with or without heat treatment as a form of an endosteal implant in the calvarium of sheep. The experiment proceeded with the accomplished fact that the mechanical strength increases and the degradation rate decreases when post-processing heat treatment is performed on the WE43 alloy [131]. In their study, the WE43 implant without heat treatment showed a faster decomposition rate and increased bone remodeling and gas pocket formation compared to the WE43 implant with heat treatment, and it was concluded that WE43 alloy with heat treatment seems to show better biological behavior [130].

### 3.5. Clinical Application

To the best of our knowledge, there is only one case report using Mg-based osteofixative materials in the maxillofacial area in humans. Leonhardt et al. [132] presented a case report in 2017, using absorbable Mg-based headless screws in five patients with fractures of the condylar head. Adequate reduction of the fractures and favorable screw positioning were observed through postoperative cone-beam computed tomography. No limitation of mandibular movement was observed within three months post-operation, and all the patients showed satisfactory occlusion. Moreover, all patients showed neither edema associated with hydrogen gas nor other complications due to screw degradation. Although the cost problem (five times that of the conventional titanium screw) remains, Mg screws showed excellent biocompatibility and similar results to the titanium screws. Hence, they demonstrated that further clinical studies with a longer follow-up were needed to apply Mg screw in the clinical situation. Currently, there are no randomized controlled trials or cohort studies using Mg-based plates and screws in maxillofacial surgery; however, this will be possible in the future if biocompatibility is further verified through additional in vivo studies.

## 4. Conclusions

Bioabsorbable plates and screws are applied clinically for various purposes, and they were developed to replace conventional titanium plates and screws. The polymer-based osteofixation systems in current use show clinically satisfactory grades. As a result, they are widely used in surgeries that may require secondary surgery for removal of plates and screws such as orthognathic surgery and fixation of pediatric maxillofacial fractures. However, there remain several limitations, such as concerns about their physical strength, the hassle of bending and molding processes of plates, and radiolucent images that interfere with postoperative monitoring to overcome. Mg is in the spotlight as a new osteofixative material, and its absorbability, as well as superior mechanical strength and favorable biocompatibility, was demonstrated in several recent studies. However, it still requires additional verification for use in the human body.

Further future development of polymer-based materials should focus on the improvement of mechanical strength, reduction of foreign-body reactions, controllability of bioabsorption rate, and improved operability in procedures such as plate bending and self-tapping. More randomized controlled trials or prospective studies are needed to demonstrate that various polymer-based materials are superior or comparable to titanium-based plates and screws. Regarding the future development of Mg, further research is required to enhance biocompatibility, control corrosion rates, and reduce the production of hydrogen gas. It should be possible to compare Mg-based materials of each study through the development of a set of standardized protocols for the assessment of corrosion and biocompatibility. Furthermore, more extensive collaboration with clinicians is encouraged to allow the design and development of the Mg-based materials at the earliest stages in specific clinical indications. Constant efforts and cooperation among researchers and clinicians of various fields should be made to develop superior and biocompatible osteofixative materials.

## Figures and Tables

**Figure 1 biomedicines-08-00300-f001:**
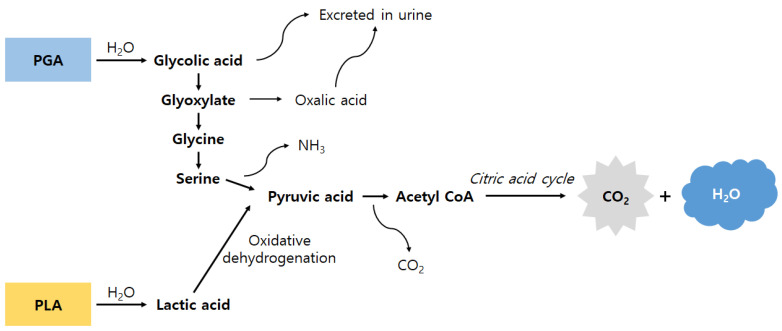
Schematic illustration showing the degradation of PGA and PLA polymers and copolymers. Glycolic acid and lactic acid monomers are produced via the hydrolytic degradation of these polymers or copolymers. Glycolic acid monomers are metabolized into glyoxylate and then converted to glycine by glycine transaminase (some of the produced glycolic acid monomers are excreted in urine). Glycine is transformed into pyruvic acid and enters the citric acid cycle. The lactic acid monomer is also oxidized to pyruvic acid and enters the citric acid cycle. As a result, carbon dioxide and water are finally produced (adapted and modified from Reference [26]).

**Figure 2 biomedicines-08-00300-f002:**
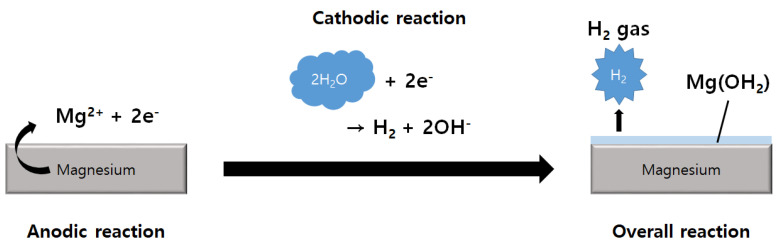
Schematic illustration showing the reactions associated with the corrosion mechanism caused when Mg is exposed to aqueous conditions.

**Table 1 biomedicines-08-00300-t001:** System of polymer-based bioabsorbable plates and screws applied for osteofixation in maxillofacial bone surgery.

Product Name	Manufacturer	Polymer Composition	Degradation Time
Biofix^®^ SR-PGA	Bionx implants, Tampere, Finland	SR-PGA	6 weeks
Biofix^®^ SR-PLLA	Bionx implants, Tampere, Finland	SR-PLLA	5–7 years
Resomer^®^ LR708	Evonik Industries, Darmstadt, Germany	PLLA (70%) + PDLLA (30%)	2–3 years
MacroPore^®^	MacroPore Biosurgery Inc., San Diego, CA, USA	PLLA (70%) + PDLLA (30%)	2–3 years
Macrosorb^®^	MacroPore Biosurgery Inc., San Diego, CA, USA	PLLA (70%) + PDLLA (30%)	2–3 years
Biosorb FX^®^	Linvatec Biomaterials Ltd., Tampere, Finland	PLLA (70%) + PDLLA (30%)	2–3 years
Resorb X^®^	KLS Martin Group, Tuttlingen, Germany	PLLA (50%) + PDLLA (50%)	12–30 months
PolyMax^®^	Synthes, Oberdorf, Switzerland	PLLA (70%) + PDLLA (30%)	2 years
PolyMax^®^ RAPID	Synthes, Oberdorf, Switzerland	PLLA (85%) + PGA (15%)	12 months
Rapidsorb^®^	DePuy Synthes, West Chester, PA, USA	PLLA (85%) + PGA (15%)	12 months
Lactosorb^®^	Lorenz, Jacksonville, FL, USA	PLLA (82%) + PGA (18%)	12 months
Delta^®^	Stryker Leibinger Corp., Kalamazoo, MI, USA	PLLA (85%), PGA (10%), PDLA (5%)	8–13 months
Inion CPS^®^	Inion Inc., Tampere, Finland	PLLA, PGA, TMC–proportion varies	2–4 years
Inion CPS^®^ baby	Inion Inc., Tampere, Finland	PLLA, PGA, TMC–proportion varies	2–3 years
Osteotrans-MX^®^	TEIJIN Medical corp., Osaka, Japan	PLLA (60–70 wt%), u-HA (30–40 wt%)	4.5–5.5 years

SR, self-reinforced; PGA, polyglycolic acid; PLLA, poly-l-lactic acid; PDLLA, poly-d-l-lactic acid; TMC, trimethylene carbonate; u-HA, unsintered hydroxyapatite; USA, United States of America.

**Table 2 biomedicines-08-00300-t002:** Magnesium (Mg) alloys studied for the application of plates and screws in maxillofacial surgery.

Metals	Chemical Composition	Tensile Strength (MPa)	Tensile Yield Strength (MPa)	Elongation (%)
Pure Mg	98.8% Mg or higher	86	20	13
Mg–Ca–Zn	Mg, 5 wt.% Ca,1 wt.% Zn	210	95	11
WE43	Mg, 3.78 wt.% Y, 2.13 wt.% Nd, 0.46 wt.% Zr	260	160	6
Extruded WE43	Mg, 3.78 wt.% Y, 2.13 wt.% Nd, 0.46 wt.% Zr	303	195	6
ZK60 *	Mg, 4.8–6.2 wt.% Zn, 0.64 wt.% Zr, 0.012 wt.% Mn, 0.0021 wt.% Fe, 0.0016 wt.% Si, 0.0014 wt.% Al, 0.001 wt.% Cu	328	251	15

Data were compiled from Byun et al. 2020 [118], Byun et al. 2020 [119], and Yang et al. 2014 [120]. * Plates were coated with poly-l-lactic acid (PLLA).

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
