# Peer review of "Bioabsorbable Osteofixation Materials for Maxillofacial Bone Surgery: A Review on Polymers and Magnesium-Based Materials"

_biomedicines, 2020, doi:10.3390/biomedicines8090300_

Round 1
Reviewer 1 Report
Good and interesting review covering the recent research regarding bioabsorbable osteofixation materials in maxillofacial bone surgery.
Author Response
We thank you for your time and consideration for our submission (Biomedicines-882767). Below we address the reviewers’ comments and list of changes that we made to our manuscript according to their reports. The original reviewers’ comments are provided black color, whereas our answers are given in red color. The appropriate changes made in the revised manuscript are provided highlighted, leaving the “track changes” function.
Point 1: Good and interesting review covering the recent research regarding bioabsorbable osteofixation materials in maxillofacial bone surgery. 

Response 1: There has been a development of changes in osteofixation materials in bone surgery. Metal screws or plates, which are frequently used, are increasingly diversified into bioabsorbable materials. We have reviewed materials that have received attention until recently. Thanks for your comment.
Reviewer 2 Report
Research is well conducted. Conclusions are in correlation with goals. Results are important for further investigations as well as for clinical work.
Author Response
We thank you for your time and consideration for our submission (Biomedicines-882767). Below we address the reviewers’ comments and list of changes that we made to our manuscript according to their reports. The original reviewers’ comments are provided black color, whereas our answers are given in red color. The appropriate changes made in the revised manuscript are provided highlighted, leaving the “track changes” function.
Point 1: Research is well conducted. Conclusions are in correlation with goals. Results are important for further investigations as well as for clinical work.

Response 1: There has been a development of changes in osteofixation materials in bone surgery. Metal screws or plates, which are frequently used, are increasingly diversified into bioabsorbable materials. We have reviewed materials that have received attention until recently. In order to accurately contain the message in the conclusion section, it was further supplemented.
Thanks for your comment.
Reviewer 3 Report
The article presented for review deals with the bioresorbable materials for osteofixation. The topics discussed are up-to-date and very important. In my opinion, such a review article is needed to collate knowledge on biodegradable materials used in osteofixation. Hovewer, despite the authors' good intentions, the presented article has many shortcomings that must be corrected before being accepted, especially in the part related to magnesium and its alloys. The article also lacks the authors' own comments on the cited results.
- The title of the manuscript suggests the bigger set of the materials described. In fact, the authors presented only 2 groups of biodegradable materials: polymers as well as Mg and its alloys. So, it is suggested to refine the second part of the title to, for example, “A Review on Polymers and Magnesium-Based Materials”
- Intruduction
Page 1 line 37: “Titanium” – do the authors mean pure titanium or its alloys as well ?
P1L38: “…plates show sufficient strength for rigid fixation …” - strength is not responsible for achieving rigidity, Young’s modulus does.
P2L4-11: no critical assessment of the chemical composition of polymers and the requirements for their purity. Are there any clinical problems related to the presence of such materials in the human body?
P2L12: “…cumbersome molding procedures…” - the authors do not mention anywhere about the possibility of producing the elements for osteosynthesis by additional manufacturing
P2L19-20: The purpose should be rewritten in order to avoid misunderstandings regarding the number of material groups discussed. Moreover, in this sentence there are repetitions of the word "review" and “overwiev”
- Chapter 2: Conventional bioabsorbable plates and screws
P2L23: I suggest to start from the second sentence (historically)
P2L30: self-reinforced (SR-) – please, explain the mechanism of self-reinforcement
Table 1 should be completed: please add the mechanical properties (if available) and sources of the data in the following columns. Please, also add the mechanical requirements for such materials (in the table or in the paragraph).
P3L5: section 2.1. “Characteristic of the polymer-based…” rather than “Type of the polymer based…”
P3L7: “…has been used clinically for the first time” – does it mean, it was the first material used ? I strongly recommend to use the values of mechanical properties. It is not enough to mention "high strength", because in the case of polymers it is much lower than for most metals.
P3L15: “PLA is another bioabsorbable polymer with a high-molecular-weight” – add the value.
P3L17: “PLLA has been regarded as the first generation…” – I suggest to use as the first sentence in this section. The same suggestion (generations) for the following sections.
P3L19: “…and therefore is resistant to hydrolysis” – is it completely resistant ?
P4L2: “insufficient strength” – please, clarify
P4L7: “…particles may cause some inflammatory response” [reference needed].
Figure 1: I suggest completing the degradation scheme, especially at the sites of transformation of PGA into Glycolic acid and PLA into Lactic acid and then into Pyruvic acid (similar to the CO2 release in the following steps shown)
Fig. 1 Caption: “Schematic illustration showing the degradation of PGA and PLA polymers…” rather than “Schematic illustration showing the degradation of PGA, PLA polymers…”. I also suggest divide the long sentence in Line 27 into 2 shorter ones.
Subsection 2.1.4. Suggested title change, the authors use abbreviations only
P4L32-33: u-HA - why is unsintered HA used instead of sintered? Please, explain
P5L1: “…composite materials show higher mechanical strength…” – higher than what ?
P5L8: “…while the biomechanical stress exposed to this region is low” – I think, relatively low.
P5L23: “…reported delayed complications associated with the material [23].” – what kind of complications ?
P5L26 and 33: it is recommended to insert reference number right after the name of cited Author.
P5L40: “Le Fort I…” – use the abbreviation right after the designation
P6L5: “Although immediate loosening of the screw occurred in 3 patients…” – is there any information on the causes of the loosening?
P6L11: “…absorbable device in 2 patients.,” remove comma
P6L25: “…the failure of fixation” – for what reason ?
P6L25: “Food and Drug Administration” – what country ?
P6L31 and 36 - standardize the abbreviation “LFI”
P6L33, 38 and 49 – expected Authors' comment on failure cases
P7L12: “…although it could not be told which polymer is better.” – expected Authors' comment
P7L16-17: “…physical strength, questions about their ability to endure high loads…” – is there any difference between “physical strength” and “ability to endure high loads”?
P7L30: the word "system" used twice
P7L30-31: “…the necessity of removing the absorbable plates and screws due to complications such as infection was found to be twice as high as that of titanium” – a comment is expected
P7L34: “Furthermore, in another meta-analysis…” move the reference [63] here
P7L37: what do the authors mean by “stronger strengths”
Chapter 3
I suggest change the title in accordance with Chapter 2. At least one sentence is needed in the beginning of this chapter as the introduction.
Do not start the sentence with an abbreviation (Mg).
P7L40-50: In this paragraph, the authors presented the role of magnesium in the human body. How does this relate to the "high" magnesium content in the vicinity of the osteofixation elements?
P8L8: “…the corrosion of Mg and investigate the biocompatibility…” - if Mg corrodes and induces metallosis, biocompatibility is difficult. Authors comment required.
A table listing the mechanical properties of magnesium and common magnesium alloys is required, together with their chemical composition and degradation time, similar to Table 1.
P8L22: It is suggested to move the sentence “The application of Mg for orthopedic use showed slightly more varied results.” to the next paragraph.
P8L26: remove “resulted from the use of two different metals” which is obvious
P8L28-29: “…after three months when used alone, and wholly corroded after 7-10 months” the sentence should be rewritten to clarify
P8L32 and 34: move the reference numbers right after the names of the cited Authors
P8L46: “environment” rather than “conditions”
P8L49 and next: correctly write down chemical equations with subscripts
Fig 2: The black arrows suggest the changes occurring only in magnesium. Please redraw the image to make it unequivocal
Paragraph starting from the line 6 to 16 need to be rewritten. The information in it is school based and adds nothing to the review.
Title of the section 3.3: “Methods for improving the limitation of Mg for biomedical application” - do the authors mean an increase in limitation ? It doesn’t make sense, I think.
P9L18-20: “Various techniques are currently being studied to improve the appropriate mechanical properties and corrosion resistance of Mg as a biomaterial, along with the biocompatibility and potential osseointegration ability.” rather than “Various techniques are currently being studied to improve the appropriate mechanical properties and corrosion resistance, along with the biocompatibility and potential osseointegration ability of Mg as a biomaterial.”
The second paragraph of the 3.3 Section (lines 22-33) contains very general information. Please describe in detail, including the role of elements in Mg alloys, applied heat treatment and obtained microstructures, on this basis evaluate the corrosion resistance. Also, elements such as Cadmium can be toxic - please comment. Furthermore, Lithium is an element even more active than magnesium - authors' comment is required regarding the possibility of using Mg-Li alloys as biomaterials.
P9L37-38 – definition of surface modification and its methods are commonly known. Please specify in detail what kind of surface modification is used for Mg and its alloys for biomaterials applications.
Section 3.4: P10L15 to P12L15: this section is too general, it provides only basic information. It all should be removed. Please focus on the topic of the section and describe it well.
P12L22-28: FEI simulation is as good as the model allows. This paragraph adds only a little to the topic of the section.
P12L30-31: “…Mg alloy WE43 with or without plasma electrolytic surface coating…” – lack of chemical composition of the alloy and its characteristics, also no information on kind of surface coating. The same for ZK60 (P13L13)
P12L40-42: sentences should be rewritten
P12L45: “…had no significant effect on bone healing, biocompatibility, and corrosion kinetics…” – biocompatibility and corrosion, the same as above (P8L8)
P13L6: “…it decreased spontaneously…” – why? any comment ?
P13L9: “…increased mechanical strength through the extrusion process” - the process itself does not change the properties of the material. Properties result from changes in the material caused by the process. It is necessary to comment, what happens in the material as a result of the extrusion process.
P13L10-11: “In addition, it was observed that the screw fixed in the bone showed a slower absorption rate compared to the plate separated from the bone.” – it is obvious. I strongly recommend the Authors take a more critical look at the cited content.
P13L21-23: “The experiment proceeded with the accomplished fact that the mechanical strength increases, and the degradation rate is improved when post-processing heat treatment is performed on the WE43 alloy” - what the authors mean by "improved degradation rate"? A Comment is expected on the influence of heat treatment on mechanical properties and corrosion resistance.
The “Future development” section is desirable before Conclusions section
Chapter 4 – Conclusions – please consider improving this chapter, taking into account the purpose of the work, advantages and disadvantages of the described materials and indications for the future.
References
references 121, 122, 138 should be corrected
Round 2
Reviewer 3 Report
The corrected article is significantly improved, however, in my opinion, unsatisfactory yet.
The review article should present a coherent whole. The topic concerns bioabsorbable materials for osteofixation. Therefore, in addition to discussing clinical topics, it is required to characterize the materials and their basic properties. Without such characteristics, the article should not be published.
The part on magnesium-based materials is still not consistent with the "polymer" part. Furthermore, the review article should be more critical and show different points of view or at least the Author’s comments. Biomedicines journal, with such a high IF, requires review articles with high substantive value, indicating the expert and interdisciplinary knowledge of the Authors.
The following comments to the answers should be considered before publishing the article:
- “…plates show sufficient strength for rigid fixation …” – Strength is an engineering property, which is not responsible itself for achieving rigidity. If the fixation must be rigid, the plate should exhibit both: the required strength and stiffness, to prevent micro-movements. For example, polymers exhibit very low value of Young’s modulus which results in higher deformations of plate in relation to the bone. So, I suggest to use the phrase “plates show sufficient strength and Young’s modulus values for rigid fixation” instead of “plates show sufficient strength for rigid fixation”.
- Table 1. The table shows the degradation time as the only useful information. I understand that the Authors wanted to present materials for clinicians, but the use of a specific material is not arbitrary. Most often it results from the area of application and the required mechanical properties. I suggest to present at least the ranges of basic mechanical properties (tensile strength and Young's modulus) so that the reader can relate these values to, for example, titanium plates or those made of magnesium alloys. Of course, supplementing with appropriate sources is advisable.
- u-HA
I expected an explanation of the use of u-HA in this paragraph.
Moreover, the Authors should include almost all their answers in the text, which would significantly improve the quality of the article. The same for comments No. 26, 28, 32 and 52
- “higher strength” instead of “stronger physical strengths”. It seems that Authors are not engineers. It would be advisable to consult the text by an engineer or invite him as the co-author
- A table listing the mechanical properties of magnesium and common magnesium alloys is required, together with their chemical composition and degradation time, similar to Table 1.
I still don’t understand the reasons why the Authors did not include the Table on magnesium and its alloys similarly as Table 1 for polymers. This makes the article inconsistent. Aside from clinicians as potential recipients of the article, it is necessary to show the properties of these materials. This enables the reader to evaluate the properties of the materials compared to the other. One of the articles presenting such information is: Hendra Hermawan, Updates on the research and development of absorbable metals for biomedical applications. Prog Biomater. 2018 Jun; 7: 93–110 https://dx.doi.org/10.1007%2Fs40204-018-0091-4
- Section 3.4: P10L15 to P12L15
I do not agree with the authors' opinion, which is inconsistent. While the corrosion of magnesium alloys may be unknown to clinicians, biocompatibility studies are certainly familiar to them. Therefore, I stand by my decision to remove the part of methods of biocompatibility testing.
The characteristics of corrosion tests on Mg-based materials are certainly interesting, but in the present state, there is an chaos in it making it difficult to understand. The Authors should organize this fragment and specify which corrosion processes are dominant and what is the role of alloying elements in these processes instead of describing the methods. If the readers do not understand the corrosion processes, the description of the methods will be all the more useless.
In the “polymer” part, the Authors have included a subsection 2.2. “Clinical application”. There is no such subsection in the part devoted to Mg-based materials. If it were separated from the present text, it is more than 2 times smaller than the corresponding part in polymers section. Please consider expanding this part.
Round 3
Reviewer 3 Report
The manuscript has been significantly revised. My main comments were taken into account. The article may now be published.